# ADAPTING INFORMATIVE STRUCTURES FOR CROSS-DOMAIN FEW-SHOT SEGMENTATION

## ABSTRACT

Cross-domain few-shot segmentation (CD-FSS) aims to segment objects of novel classes under domain shifts, using only a few mask-annotated support samples. However, directly applying pretrained CD-FSS models to unseen domains is often suboptimal due to their limited coverage of domain diversity by fixed parameters trained on source domains. Moreover, simply adjusting hand-selected model parameters, such as test-time training, typically neglects the distinct domain gaps and characteristics of target domains. To address these issues, we propose adapting informative model structures for target domains by learning domain characteristics from few-shot labeled support samples during inference. Specifically, we first adaptively identify domain-specific model structures by measuring parameter importance using a novel structure Fisher score in a data-dependent manner. Then, we progressively train the selected informative model structures with hierarchically constructed training samples, progressing from fewer to more support shots. Our method selectively and gradually adapts the model to target domains, optimizing model adaptation, minimizing overfitting risks, and maximizing the use of limited support data. The resulting Informative Structure Adaptation (ISA) method effectively addresses domain shifts and equips existing few-shot segmentation models with flexible adaptation capabilities for new domains, eliminating the need to re-design or retrain CD-FSS models on base data. Extensive experiments validate the effectiveness of our method, demonstrating superior performance across multiple CD-FSS benchmarks.

## 1 INTRODUCTION

Few-shot semantic segmentation (FSS) aims to segment novel classes using a limited number of support samples. It typically trains a conventional support-query matching network to transfer class-agnostic patterns from extensive base data to novel classes. Existing FSS methods (Fan et al., 2022a; Nguyen & Todorovic, 2019; Lu et al., 2021; Zhang et al., 2019b) have made significant progress on in-domain class generalization due to various well-designed matching and training techniques.

Despite their success, existing few-shot segmentation methods often struggle with domain shifts, particularly when test and training data come from different domain distributions. This challenge underscores the significance of cross-domain few-shot segmentation (CD-FSS), which aims to generalize to new classes and unseen domains using minimal annotated data from the target domain.

Existing CD-FSS methods (Lei et al., 2022; Su et al., 2024a; Wang et al., 2022b; Huang et al., 2023) typically train models by leveraging abundant base data from the source domain and then directly applying the trained models to various target domains. However, the issue arises because CD-FSS models are trained on limited domain data with frozen parameters, while the potential target domains can be diverse and arbitrary. Therefore, it is necessary to adapt the trained models to target domains during inference by utilizing few-shot labeled support samples.

Test-time training (TTT) (Wang et al., 2020; Sun et al., 2020b) effectively adapts models to target domains by learning from test data. Although promising, most existing TTT methods adjust the same manually-selected trainable structures across different domains, disregarding the distinct domain gaps and characteristics of the target domains. For instance, in a domain-agnostic manner, MCS (Liang et al., 2019) and Tent (Wang et al., 2020) respectively train the classifier and transformation parameters across all possible target domains. However, different domains, or even individual test images,

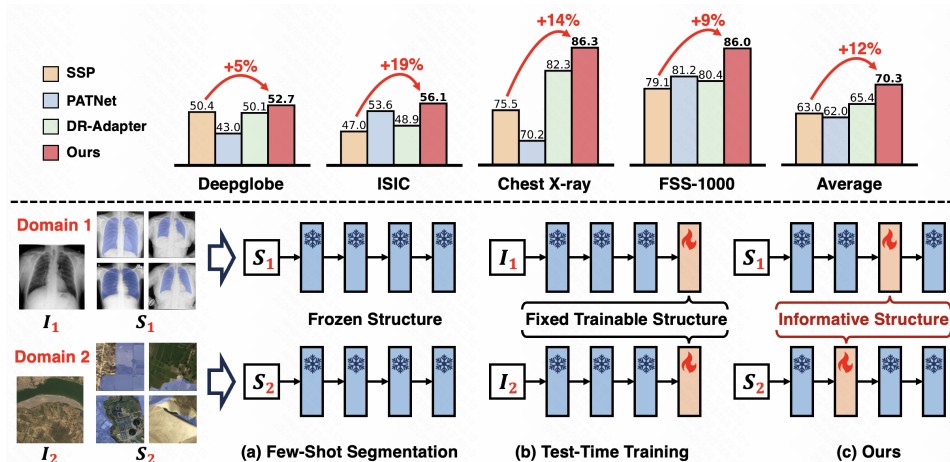

Figure 1: **Top:** Performance comparison of various methods on 5-shot CD-FSS tasks across four datasets. Our ISA method outperforms all other approaches, including FSS (SSP), CD-FSS with TTT (PATNet), and CD-FSS without TTT (DR-Adapter). **Bottom:** Comparison of related methods. (a) Few-Shot Segmentation: Directly apply a frozen model structure across all domains without adaptation. (b) Test-Time Training: Simply fine-tunes a fixed, trainable model structure, such as the last layer, across all domains. (c) Ours (ISA): Fine-tunes the selected informative model structure, which varies across different domains.

exhibit distinct properties related to various model structures, such as specific model layers. For example, the satellite images in DeepGlobe (Demir et al., 2018) dataset primarily rely on low-level texture analysis for parsing structured and detailed remote sensing areas. In contrast, Chest X-ray images (Candemir et al., 2013) typically require middle- and high-level semantic understanding to distinguish pneumonia-affected areas from normal lung regions in medical image analysis. Thus, dynamically selecting trainable model structures is crucial for adapting to various target domains with distinct characteristics.

To address these challenges, we propose a novel method, *Informative Structure Adaptation (ISA)*, specifically designed for cross-domain few-shot segmentation. We explore methods for *identifying* and *adapting* domain-specific informative structures during inference by learning domain characteristics from few-shot labeled support samples. Given the varying reliance on model structures across different domains (Yosinski et al., 2014; Liang et al., 2019), the *Informative Structure Identification (ISI)* module identifies domain-sensitive model structures by measuring parameter importance in a data-dependent manner. First, we compute empirical Fisher information to reduce computational overhead, inspired by the parameter importance metric used in continual learning (Kirkpatrick et al., 2017). Building on this, we propose a novel structure Fisher score to guide the identification of informative model structures. This strategy optimizes model adaptation for varying domain characteristics and mitigates the risk of overfitting in few-shot scenarios.

Once informative model structures are obtained, optimizing trainable parameters becomes essential, particularly in few-shot scenarios. Conventional test-time training methods typically optimize the model using a single test image that shares the same class space as the training data. In contrast, CD-FSS requires simultaneous generalization to new classes and unseen target domains, utilizing multiple labeled support images. Therefore, it is essential to fully utilize few-shot support data to tackle the challenges of class and domain generalization. We propose a novel *Progressive Structure Adaptation (PSA)* module that trains the model with hierarchically constructed training samples, progressing from fewer to more support shots. Specifically, we initially create support-query training pairs by cyclically designating each support image as the pseudo query image. Subsequently, we extend the training pairs by progressively increasing the number of support shots. This strategy enables the model to gradually adapt to domain shifts and maximizes the use of limited support data during inference for the challenging CD-FSS.

Our approach fundamentally differs from conventional few-shot segmentation and test-time training methods, as shown in Figure 1. We are the first to leverage few-shot annotated support data to adaptively identify and progressively adapt informative model structures during test-time training for

CD-FSS. In contrast, most prevailing test-time training methods (Wang et al., 2020; Su et al., 2022; Wang et al., 2022a; Su et al., 2024b) manually define trainable model structures and directly train models on accessible test data, which are proven suboptimal by our empirical analysis. Moreover, most test-time training methods fail to generalize to novel classes and semantic segmentation tasks. Although PATNet (Lei et al., 2022) introduces a test-time training method for CD-FSS, it is specifically designed to train fixed anchor layers and requires pre-training on the source domain. Therefore, applying PATNet to other few-shot semantic segmentation methods is non-trivial. In contrast, our method is model-agnostic, requires no additional learnable parameters, and can easily equip existing few-shot segmentation models with flexible adaptation capabilities for new domains, eliminating the need to redesign or retrain CD-FSS models on base data. In summary, our key contributions include:

- We introduce a novel Informative Structure Adaptation (ISA) method that adaptively identifies and progressively adjusts informative model structures during inference for CD-FSS.

- The Informative Structure Identification (ISI) module dynamically identifies domain-sensitive model structures in a data-dependent manner, while the Progressive Structure Adaptation (PSA) module progressively addresses domain shifts by adapting the model with an increasing number of support shots.

- Our ISA generalizes effectively across multiple unseen target domains and is remarkably simple, enabling the adaptation of existing few-shot segmentation methods for CD-FSS without the need for redesigning or retraining CD-FSS models on base data.

## 2 RELATED WORKS

**Few-Shot Semantic Segmentation.** Few-shot semantic segmentation, pioneered by Shaban et al. (Shaban et al., 2017), aims to predict dense masks for objects of novel classes using only a limited number of labeled support images. The mainstream prototype-based methods (Dong & Xing, 2018; Li et al., 2021; Wang et al., 2019) perform segmentation by measuring the similarity between the query features and representative support prototypes incorporating various improvements (Siam et al., 2020; Liu et al., 2020; Zhang et al., 2021a; Zhuge & Shen, 2021). The affinity-based methods (Lu et al., 2021; Zhang et al., 2021b; Peng et al., 2023; Min et al., 2021; Tian et al., 2020) establish detailed dense correspondence between query and support features through feature concatenation and leverage a learnable CNN or transformer module for segmentation prediction. Recently, foundation models like SAM (Kirillov et al., 2023) present a novel opportunity for few-shot segmentation (Liu et al., 2024; Zhang et al., 2024a), due to their remarkable transfer capability on tasks and data distributions beyond the training scope. However, these methods do not consider the domain shifts problem, leading to poor generalization performance when encountering new domains during testing.

**Cross-Domain Few-Shot Semantic Segmentation.** Cross-domain few-shot semantic segmentation has recently received increasing attention. PATNet (Lei et al., 2022) introduces a feature transformation layer that seamlessly maps query and support features across diverse domains into a unified feature space, effectively tackling the intra-domain knowledge preservation issue in CD-FSS. RD (Wang et al., 2022b) utilizes a memory bank to reinstill meta-knowledge from the source domain, thereby improving generalization performance in the target domain. Subsequently, DARNet (Fan et al., 2023) and RestNet (Huang et al., 2023) approach the problem from distinct perspectives, focusing on bridging domain gaps through dynamic adaptation refinement and knowledge transfer, respectively. Inspired by these pioneering efforts, PMNET (Chen et al., 2024a) presents a comprehensive solution capable of addressing both in-domain and cross-domain FSS tasks concurrently by capturing pixel relations within each support-query pair. Unlike previous CD-FSS approaches, our method effectively addresses domain shifts in CD-FSS during inference by adaptively identifying and gradually adapting informative model structures on few-shot annotated support samples, and is capable of seamlessly adapting current few-shot segmentation methods to address domain shifts problem.

**Test-Time Training.** Normally, once a well-trained model is deployed, it remains static without further alterations. In contrast, test-time training (TTT) (Sun et al., 2020a) adapts models to the deployment scenario by leveraging unlabeled data available at test time. The mainstream self-supervised learning-based methods (Liu et al., 2021b; Wang et al., 2021; Liang et al., 2020; Goyal et al., 2022; Gandelsman et al., 2022) leverage the available unlabeled test data to facilitate model

adaptation to the target domain using self-supervised learning techniques. The feature alignment-based methods (Su et al., 2022; Jung et al., 2023; Wang et al., 2023a) attempt to rectify the feature representations for the target domain. Some works attempt to apply TTT to address the semantic segmentation problem. For instance, MM-TTA (Shin et al., 2022) utilizes multiple modalities to provide reciprocal TTT self-supervision for 3D semantic segmentation. Similarly, CD-TTA (Song et al., 2022) explores domain-specific TTT for urban scene segmentation using an online clustering algorithm. OCL (Zhang et al., 2024b) proposes an output contrastive loss to stabilize the TTT adaptation process for extreme class imbalance and complex decision spaces in semantic segmentation. These methods typically adjust fixed hand-selected trainable structures on one single test image for different domains. In contrast, our method dynamically adapt domain-specific informative structures by learning domain characteristics from few-shot labeled support samples.

## 3 METHOD

Cross-Domain Few-Shot Segmentation (CD-FSS) aims to transfer knowledge learned from the source domain to new categories in unseen target domains using minimal annotated support samples. The model is typically trained on the source domain and then evaluated on target domains, ensuring no label space overlap between the source and target domains.

### 3.1 BASELINE METHODS

**Few-Shot Segmentation Model.** Mainstream few-shot semantic segmentation model can be formulated as follows: The input support and query images $\{I_s, I_q\}$ are processed by a weight-shared backbone to extract image features $\{\mathcal{F}_s, \mathcal{F}_q\}$:

$$\mathcal{F}_s = f(I_s; \theta), \mathcal{F}_q = f(I_q; \theta), \tag{1}$$

where $f$ denotes the image encoder with parameters $\theta$. Then, the support features $\mathcal{F}_s$ and groundtruth masks $\mathcal{M}_s$ are fed into the masked average pooling layer (MAP) to generate support prototypes $\mathcal{P}_s$. The final prediction is made by measuring the cosine similarity between $\mathcal{P}_s$ and $\mathcal{F}_q$.

**Model Structure Adaptation Baseline.** Our model structure adaptation baseline for CD-FSS is derived from the Test-Time Training (TTT) method, which typically adapts the pre-trained source domain model during evaluation using the available test data. In CD-FSS, we segment the unlabeled query image using the few-shot support set $S = \left\{ \left( I_s^i, \mathcal{M}_s^i \right) \right\}_{i=1}^K$ containing $K$ support images with groundtruth masks. We leverage the labeled support data to train the few-shot matching model for adapting model structures by constructing support-query pairs with mask labels. Specifically, we randomly select one support data $S_q^i = (I_s^i, \mathcal{M}_s^i)$ as the pseudo query data, and treat the remaining support samples as a new support set $S \setminus S_q^i$, creating a support-query training pair $(S \setminus S_q^i, S_q^i)$. Then, we extract support prototypes $\mathcal{P}_s^i$ and query features $\mathcal{F}_q^i$ for test-time training:

$$\mathcal{L}_{\mathrm{T}}^i = BCE \left( \mathrm{cosine} \left( \mathcal{P}_s^i, \mathcal{F}_q^i \right), \mathcal{M}_q^i \right), \tag{2}$$

where BCE is the binary cross entropy loss and $\mathcal{M}_q$ is the groundtruth mask of the pseudo query image. Eventually, we train the model by optimizing the loss: $\theta^* = \arg\min_\theta \mathcal{L}_{\mathrm{T}}^i(\mathcal{P}_s^i, \mathcal{F}_q^i, \mathcal{M}_q^i; \theta)$, where $\theta$ denotes the trainable parameters of the model and $\theta^*$ denotes the updated model parameters after training. To prevent overfitting, we follow the common practice (Boudiaf et al., 2021; He et al., 2020) to train only the final convolutional layer of the model during test-time training.

### 3.2 INFORMATIVE STRUCTURE IDENTIFICATION

To adapt the model to varying domain characteristics, we first investigate *how to identify* domain-specific informative structures from few-shot labeled support samples during inference, rather than manually defining trainable model layers.

**Fisher Information Matrix** (FIM) can evaluate the significance of parameters concerning a specific task and data distribution. Given a model with parameters $\theta$, input $x_i$, output $y_i$ and output probability $p_\theta(y_i|x_i)$, the FIM can be computed as $F_\theta = \mathbb{E}_{x \sim p(x)} \left[ \mathbb{E}_{y \sim p_\theta(y|x)} \left( \frac{\partial \log p_\theta(y|x)}{\partial \theta} \right) \left( \frac{\partial \log p_\theta(y|x)}{\partial \theta} \right)^\top \right]$.

The matrix $F_\theta \in \mathbb{R}^{|\theta| \times |\theta|}$ can alternatively be understood as representing the covariance of the gradients of the log likelihood with respect to the parameters $\theta$.

**Empirical Fisher Information.** However, directly computing the Fisher Information Matrix for the pre-trained CD-FSS model involves significant computational overhead due to the $|\theta| \times |\theta|$ scale. Thus, we simplify FIM for CD-FSS, inspired by the parameter importance metric used in continual learning (Kirkpatrick et al., 2017). Specifically, we concentrate on the support samples $K$ in the target domain and utilize the diagonal elements of the "Empirical Fisher" to evaluate the importance of the pre-trained model parameters for cross-domain tasks. Specifically, for the $l$-th convolutional layer, we derive the empirical Fisher information of its $u$-th trainable parameters as $F_{\theta_{l,u}} = \frac{1}{|K|} \sum_{j=1}^{|K|} \left( \frac{\partial \log p_\theta(y_j|x_j)}{\partial \theta_{l,u}} \right)^2$. Correspondingly, a relatively large value of $F_{\theta_{l,u}}$ indicates that the parameter $\theta_{l,u}$ is crucial for the cross-domain task.

**Structure Fisher Score.** Now, we can compute the empirical Fisher information for all parameters of the model based on labeled support samples. We observe that the empirical Fisher information is typically distributed sparsely throughout the model, with many low-value entries in each convolutional layer. Directly fine-tuning the most sensitive unstructured parameters may lack the representational capacity to handle severe domain shifts. Therefore, we propose identifying informative model structures, *i.e.*, convolutional layers, for subsequent model adaptation. Specifically, we compute the maximum absolute value of empirical Fisher information across all $U$ trainable parameters within the $l$-th layer as its structure fisher score $F_{\theta_l}^*$:

$$F_{\theta_l}^* = \max \left( |F_{\theta_{l,1}}|, |F_{\theta_{l,2}}|, \ldots, |F_{\theta_{l,u}}|, \ldots, |F_{\theta_{l,U}}| \right). \tag{3}$$

Model layers with higher structure Fisher scores are typically more important for model training (Liu et al., 2021a) because of their greater contribution to the optimization process. Updating only the informative model structures preserves the model's ability to fit few-shot data and regularizes training to mitigate the risk of overfitting. Therefore, we select the model layer with the highest structure Fisher score and update its parameters $\theta_{\tt tr}$ for model structure adaptation during inference, while freezing all other parameters to minimize the risk of overfitting in few-shot scenarios:

$$\theta_{\tt tr} = \theta_{l^*}, \text{ where } l^* = \arg\max_l \{F_{\theta_l}^*\}. \tag{4}$$

### 3.3 Progressive Structure Adaptation

After identifying informative model structures, it is essential to optimize trainable parameters during inference by fully utilizing few-shot support data to tackle the challenges of CD-FSS. Therefore, we propose a Progressive Structure Adaptation (PSA) module to assist the model gradually address domain shifts by adapting informative model structures on hierarchically constructed training samples, progressing from fewer to more support shots.

**Hierarchical Training Sample Construction.** We begin by enhancing the utilization of few-shot support data, cyclically designating each support image as a pseudo query image to generate multiple support-query training pairs $\{(S \setminus S_q^i, S_q^i)\}_{i=1}^K$. To conserve computational resources, we first extract features from all support images, then compute the losses for each constructed training pair based on these features, and finally perform back-propagation to update model parameters using the averaged loss $\frac{1}{K} \sum_{i=1}^K \mathcal{L}_{\tt T}^i$ across all training pairs. Next, we vary the number of support shots from 1 to $K-1$ to construct hierarchical training pairs. Specifically, for each support shot number $n$ ($n \leq K-1$), we construct training pairs containing $n$ support samples. Therefore, we progressively increase the number of support samples to construct training pairs, optimizing the utilization of few-shot support data and enabling the model to gradually handle domain shifts during inference.

**Progressive Structure Adaptation.** To better address domain shifts in cross-domain tasks during test-time training, we propose a Progressive Structure Adaptation (PSA) module that trains the model using HTC-constructed hierarchical training pairs. The PSA method progressively trains the model with an increasing number of support shots from 1 to $K-1$, gradually reducing the domain gap. Specifically, for the support shot number $n$, the training loss is:

$$\mathcal{L}_{\text{PSA},n} = \frac{1}{K \cdot S_n} \sum_{i=1}^K \sum_{s_n} BCE \left( \text{cosine} \left( \mathcal{P}_{s_n}^i, \mathcal{F}_q^i \right), \mathcal{M}_q^i \right), \tag{5}$$

---

**Algorithm 1** Informative Structure Adaptation (ISA)

---

1: **Require:** $K$-shot support samples, and well-trained FSS model $f$.
2: Select trainable layer parameters $\theta_{\mathtt{tr}}$ using **ISI** module                    ▷ See Section 3.2
3: **for** $n$ from 1 to $K-1$ **do**                    ▷ **PSA with increasing support shots.** See Eqn. 7
4:     Extract features $\mathcal{F}$ for all samples using the updated model $f$                    ▷ See Eqn. 1
5:     **for** $i$ from 1 to $K$ **do**                    ▷ **HTC with cyclic pseudo query**
6:         Compute loss $\mathcal{L}_{\mathtt{T}}^i$ on $\mathcal{F}$ for the $i$-th query with $n$ support samples                    ▷ See Eqn. 2
7:         (Omit the computation on combinations of $n$ support samples for clarity)
8:     **end for**
9:     Compute average loss $\mathcal{L}_{\mathtt{PSA},n}$ for support shots $n$                    ▷ See Eqn. 5
10:     Back-propagation for model $f$ with $\mathcal{L}_{\mathtt{PSA},n}$
11:     Update model $f$: $\theta_{\mathtt{tr},n-1}^* \rightarrow \theta_{\mathtt{tr},n}^*$                    ▷ See Eqn. 6
12: **end for**

---

where $s_n$ denotes the $s_n$-th combination of $n$ support samples, enumerated from the $S \setminus S_q^i$ support set, with a total of $S_n$ combinations. The model parameters are updated by optimizing the loss:

$$\theta_n^* = \arg\min_{\theta_{n-1}^*} \mathcal{L}_{\mathtt{PSA},n}(\mathcal{P}_{s_n}^i, \mathcal{F}_q^i, \mathcal{M}_q^i; \theta_{n-1}^*), \tag{6}$$

where $\theta_{n-1}^*$ and $\theta_n^*$ denote the updated model parameters trained with HTC-constructed training samples with $n-1$ and $n$ support shots, respectively. Specifically, when $n=1$, the $\theta_{n-1}^*$ represents the original parameters $\theta$ of the FSS model.

Consequently, we progressively update the model parameters by gradually increasing the number of support shots from 1 to $K-1$, as follows: $\theta_1^* \rightarrow \theta_2^* \rightarrow \cdots \rightarrow \theta_n^* \rightarrow \cdots \rightarrow \theta_{K-1}^*$. This approach optimizes the use of limited support data to progressively mitigate domain shifts during inference on the support set. Hence, we term it as Progressive Structure Adaptation (PSA) module.

### 3.4 Informative Structure Adaptation

We incorporate the proposed ISI and PSA modules into the Informative Structure Adaptation (ISA) method, as shown in Algorithm 1. Specifically, we first use ISI to select the trainable parameters $\theta_{\mathtt{tr}}$ for test-time training. Then, we use PSA with hierarchically constructed training pairs to train the selected model parameters by gradually increasing the number of support shots $n$ from 1 to $K-1$:

$$\theta_{\mathtt{tr},1}^* \rightarrow \theta_{\mathtt{tr},2}^* \rightarrow \cdots \rightarrow \theta_{\mathtt{tr},n}^* \rightarrow \ldots \theta_{\mathtt{tr},K-1}^*. \tag{7}$$

Notably, unlike conventional online TTT settings, our method isolates model training among testing episodes, thereby safeguarding against data leakage and ensuring fidelity to the few-shot setting.

## 4 Experiments

We adopt the popular few-shot semantic segmentation model SSP (Fan et al., 2022a) as our baseline, trained on Pascal VOC (Everingham et al., 2010) source domain dataset. We directly apply our method to the public released, well-trained SSP model with a ResNet-50 (He et al., 2016) backbone, without any re-training on the source domain dataset. For test-time training, we use the SGD optimizer with a learning rate of 1e-3 and one training iteration to update the trainable model parameters. Following previous works (Lei et al., 2022; Su et al., 2024a), we evaluate all methods on four datasets with distinct domain shifts: Deepglobe (Demir et al., 2018) for satellite images with seven categories, ISIC2018 (Codella et al., 2019; Tschandl et al., 2018) for medical images with three types of skin lesions, Chest X-Ray (Candemir et al., 2013; Jaeger et al., 2013) for medical screening images, and FSS-1000 (Li et al., 2020) for 1000-class daily objects. The input images are resized to $400 \times 400$ pixels. In the 1-shot setting, we apply data augmentation to support images to generate two additional support images for test-time training. We use the mean Intersection-over-Union (mIoU) for evaluation. All experiments are conducted on a Tesla V100 GPU.

Table 1: Quantitative comparison results on the CD-FSS benchmark. The models are trained on Pascal VOC source domain dataset and evaluated on four datasets with distinct domain shifts. The best results are highlighted with **bold**. The † means our reproduced results. The ‡ means using the ViT-base backbone.

| Methods | Deepglobe | | ISIC | | Chest X-ray | | FSS-1000 | | mIoU | |
|---|---|---|---|---|---|---|---|---|---|---|
| | 1-shot | 5-shot | 1-shot | 5-shot | 1-shot | 5-shot | 1-shot | 5-shot | 1-shot | 5-shot |
| PGNet (Zhang et al., 2019a) | 10.7 | 12.4 | 21.9 | 21.3 | 34.0 | 23.0 | 62.4 | 62.7 | 32.2 | 31.1 |
| PANet (Wang et al., 2019) | 36.6 | 45.4 | 25.3 | 34.0 | 57.8 | 69.3 | 69.2 | 71.7 | 47.2 | 55.1 |
| CaNet (Zhang et al., 2019b) | 22.3 | 23.1 | 25.2 | 28.2 | 28.4 | 28.6 | 70.7 | 72.0 | 36.6 | 38.0 |
| RPMMs (Yang et al., 2020) | 13.0 | 13.5 | 18.0 | 20.0 | 30.1 | 30.8 | 65.1 | 67.1 | 31.6 | 32.9 |
| PFENet (Tian et al., 2020) | 16.9 | 18.0 | 23.5 | 23.8 | 27.2 | 27.6 | 70.9 | 70.5 | 34.6 | 35.0 |
| RePRI (Boudiaf et al., 2021) | 25.0 | 27.4 | 23.3 | 26.2 | 65.1 | 65.5 | 71.0 | 74.2 | 46.1 | 48.3 |
| HSNet (Min et al., 2021) | 29.7 | 35.1 | 31.2 | 35.1 | 51.9 | 54.4 | 77.5 | 81.0 | 47.6 | 51.4 |
| SSP† (Fan et al., 2022a) | 42.3 | 50.4 | 33.0 | 47.0 | 74.9 | 75.5 | 77.1 | 79.1 | 56.8 | 63.0 |
| PATNet (Lei et al., 2022) | 37.9 | 43.0 | 41.2 | 53.6 | 66.6 | 70.2 | 78.6 | 81.2 | 56.1 | 62.0 |
| PMNet (Chen et al., 2024a) | 37.1 | 41.6 | **51.2** | 54.5 | 70.4 | 74.0 | **84.6** | **86.3** | 60.8 | 64.1 |
| ABCDFSS (Herzog, 2024) | 42.6 | 49.0 | 45.7 | 53.3 | 79.8 | 81.4 | 74.6 | 76.2 | 60.7 | 65.0 |
| APSeg‡ (He et al., 2024) | 35.9 | 40.0 | 45.4 | 54.0 | **84.1** | 84.5 | 79.7 | 81.9 | **61.3** | 65.1 |
| DR-Adapter (Su et al., 2024a) | 41.3 | 50.1 | 40.8 | 48.9 | 82.4 | 82.3 | 79.1 | 80.4 | 60.9 | 65.4 |
| Ours | 44.3 | **52.7** | 37.2 | **56.1** | 83.4 | **86.3** | 78.8 | 86.0 | 60.9 | **70.3** |

Table 2: Quantitative comparison results on SUIM dataset, where models are trained on Pascal VOC.

| | ASGNet | HSNet | SCL | RD | DAM | MMT | DR-Adapter | Ours |
|---|---|---|---|---|---|---|---|---|
| mIoU | 31.9 | 28.8 | 31.8 | 34.7 | 34.8 | 35.9 | 40.3 | **44.1** |

Figure 2: Qualitative comparisons between our method and the baseline model in the 1-way 5-shot setting across four target domain datasets. We show only one support image for clarity.

## 4.1 COMPARISON WITH STATE-OF-THE-ARTS

In Table 1, we compare our method with existing cross-domain few-shot semantic segmentation methods. Our method substantially outperforms the baseline method SSP (Fan et al., 2022a), with a 4.1/7.3 mIoU average improvement in the 1-shot/5-shot settings across all datasets. Additionally, our method surpasses previous CD-FSS SOTA methods, PMNet (Chen et al., 2024a), APSeg (He et al., 2024) and DR-Adapter (Su et al., 2024a), by a large margin in the 5-shot setting. APSeg achieves slightly better performance than our method (61.3 $v.s.$ 60.9), primarily because their ViT backbone is more powerful than our ResNet-50 backbone. Note that all other CD-FSS methods require extensive re-training on the source domain dataset to learn transferable, domain-agnostic features for domain generalization. In contrast, our method can effectively and efficiently adapt existing well-trained FSS models for segmenting objects of novel classes under domain shifts without any re-training. To further validate the effectiveness of our method, we follow RD (Wang et al., 2022b) to evaluate our method on the SUIM dataset. All models are trained on Pascal VOC dataset and evaluated on SUIM (Islam et al., 2020) dataset. Table 2 shows that our method improves the SOTA performance from 40.3 to 44.1 mIoU, beating other popular methods, including ASGNet (Li et al., 2021), HSNet (Min et al., 2021), SCL (Zhang et al., 2021a), RD (Wang et al., 2022b), DAM (Chen et al., 2024b), MMT (Wang et al., 2023b) and DR-Adapter (Su et al., 2024a). Figure 2 shows qualitative result comparisons between

Table 3: Results of ablation studies for the self-guiding test-time training method. "MSA" denotes the model structure adaptation baseline, "ISI" denotes the informative structure identification module, and "PSA" denotes the progressive structure adaptation module.

| MSA | ISI | PSA | Deepglobe | ISIC | Chest X-ray | FSS-1000 | mIoU | FPS |
|-----|-----|-----|-----------|------|-------------|----------|------|-----|
| | | | 50.4 | 47.0 | 75.5 | 79.1 | 63.0 | **16.5** |
| ✓ | | | 50.9 | 48.4 | 80.8 | 78.2 | 64.6 | 8.6 |
| ✓ | ✓ | | 51.5 | 50.6 | 81.7 | 82.1 | 66.5 | 3.2 |
| ✓ | | ✓ | **53.2** | 50.8 | 84.2 | 82.0 | 67.6 | 1.4 |
| ✓ | ✓ | ✓ | 52.7 | **56.1** | **86.3** | **86.0** | **70.3** | 1.0 |

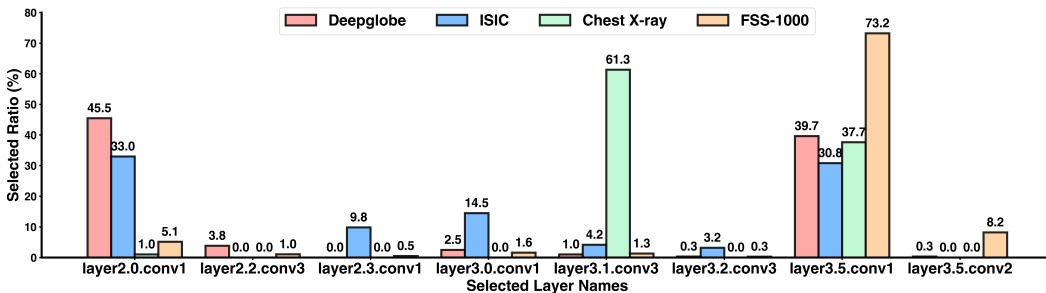

Figure 3: Selected trainable layer distribution of the informative structure identification (ISI) module.

our method and the strong baseline SSP models. By applying our ISA method, we can substantially improve the segmentation quality of the FSS method for novel objects in distinct domains.

## 4.2 ABLATION STUDIES

**Module Ablation.** In Table 3, the simple model structure adaptation (MSA) module improves the performance by 1.6 mIoU, thanks to the model adaptation for diverse target domains. The ISI module improves segmentation performance on all target domains, due to the identified informative structures for model adaptation to varying domain characteristics. The PSA module boosts the performance to 67.6 mIoU, attributing to its progressive training strategy to gradually solve domain shifts and maximal exploitation of the few-shot data. Integrating all modules, our ISA method significantly improves the performance from 63.0 to 70.3 mIoU on the strong baseline model.

**Speed Ablation.** Table 3 presents the running speed analysis of our proposed modules. Model structure adaptation (MSA) reduces the running speed from 16.5 FPS to 8.6 FPS, primarily due to the extra model forwarding step. The ISI module reduces the running speed to 3.2 FPS due to the additional model forwarding and Fisher score computation. The PSA module improves performance by 4.6 mIoU, reducing the speed to 1.4 FPS, primarily due to the multiple extra model forwarding steps required for progressive self-guiding training. Our ISA method substantially improves performance from 63.0 to 70.3 mIoU, reducing the speed to 1.0 FPS. The proposed ISA method is inherently suited for performance-demanding applications with low speed requirements, such as image annotation and offline image analysis. In Section 4.3, we further propose a fast ISA method based on our analysis, improving the running speed to 3.3 FPS while keeping competitive performance.

## 4.3 INFORMATIVE STRUCTURE ADAPTATION ANALYSIS

We conduct extensive experiments to understand our informative structure adaptation method. All experiments are performed in the 5-shot setting, focusing solely on the target module of the full ISA.

**Informative Structure Identification Mechanism.** Figure 3 summarizes the trainable layer distribution selected by ISI for various datasets. The ISI-selected trainable layers vary significantly depending on the properties of each dataset. For example, the DeepGlobe and ISIC datasets both require reliable low-level texture analysis for accurate segmentation, thus guiding the model to select more low-level trainable layers, such as "layer2.0.conv1". The FSS-1000 dataset requires high-level semantic understanding for segmenting various common objects in context, guiding the model to primarily train the high-level layers, such as "layer3.5.conv1". Figure 4 compares the training loss, testing

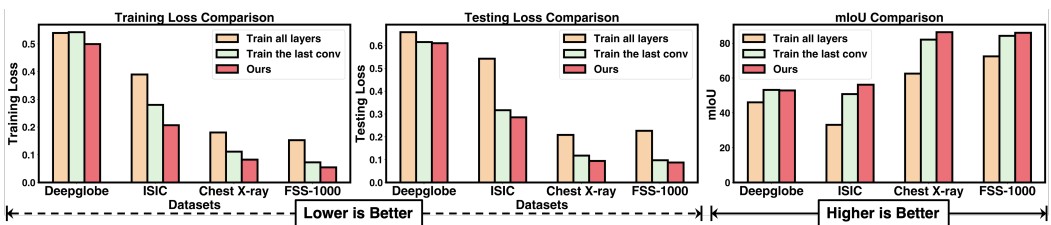

Figure 4: Comparisons on training loss, testing loss and mIoU for various model training strategies.

Table 4: Results of various hyperparameteres for informative structure identification (ISI) module.

| | Manual | ISI with different # of trainable layers | | | | | ISI with different structure Fisher scores | | | | | |
|---|---|---|---|---|---|---|---|---|---|---|---|---|
| | last conv | 1 layer | 2 layers | 3 layers | 4 layers | 5 layers | top1 | top3 | top5 | top10 | top20 | mean |
| mIoU | 67.6 | 70.3 | 70.8 | 71.0 | 70.7 | 70.6 | 70.3 | 70.2 | 70.3 | 70.3 | 70.2 | 67.8 |

Table 5: Results of using various progressive training strategies in the progressive structure adaptation (PSA) module. The "1/2/3/4" denotes training models using HTC-constructed samples with 1/2/3/4 support shots. The → denotes the sequential training procedure.

| | 1 | 2 | 3 | 4 | 1→4 | 2→4 | 3→4 | 1→2→4 | 1→3→4 | 2→3→4 | 1→2→3→4 |
|---|---|---|---|---|---|---|---|---|---|---|---|
| mIoU | 67.0 | 67.3 | 67.5 | 67.8 | 69.7 | 69.8 | 69.6 | 70.0 | 70.1 | 70.1 | 70.3 |

Table 6: Results of using various support shots for the baseline model and our method.

| | 1-shot | 3-shot | 5-shot | 8-shot | 10-shot | 15-shot | 20-shot | 30-shot |
|---|---|---|---|---|---|---|---|---|
| Baseline | 56.8 | 61.0 | 63.0 | 63.4 | 64.0 | 65.0 | 65.1 | 65.2 |
| Ours | 60.9 | 67.1 | 70.3 | 71.0 | 71.9 | 73.0 | 74.0 | 74.6 |

loss and mIoU for various model training strategies. When training all model layers, both the training loss and testing loss are significantly high across all datasets, indicating inferior generalization ability. Training only the last convolutional layer mitigates the overfitting problem. Our ISI strategy further addresses the overfitting problem, evidenced by the lowest training and testing losses, achieving the best generalization performance. This analysis validates the effectiveness and working mechanism of our selective self-guiding mechanism in addressing overfitting for test-time training in CD-FSS.

**Informative Structure Identification Hyperparameters.** Table 4 summarizes the model performance under different ISI settings. Compared to training only the last convolutional layer, our ISI substantially improves performance from 67.6 to 70.3 mIoU. By increasing the number of trainable layers, performance can be further boosted to 71.0 mIoU when ISI selects three trainable layers. We find that structure Fisher score are distributed sparsely, with many low-value scores in the convolutional parameters. Thus, the average structure Fisher score of each convolutional layer cannot represent their importance, resulting in inferior segmentation performance. In contrast, we compute the largest structure Fisher scores of each convolutional layer for trainable layer selection. We also experiment with computing the top-k largest structure Fisher scores for selecting trainable layers and achieve consistently good performance.

**Progressive Structure Adaptation Mechanism.** Our PSA module gradually addresses domain shifts using HTC-constructed training samples, progressing from fewer to more support images. As shown in Table 5, when directly trained with the 4-shot training pairs, the model performs worse than our PSA-trained model, with a 2.5 mIoU performance drop. By adding one intermediate training step $(1 \rightarrow 4, 2 \rightarrow 4, \text{or } 3 \rightarrow 4)$, the performance drop is significantly reduced to 0.5-0.7 mIoU. Adding more intermediate training steps further improves generalization performance. These results validate the importance of progressive training in gradually addressing domain shifts. Notably, our PSA does not require additional data and maximizes the use of limited support data to construct hierarchical training pairs for progressive self-guiding training.

**Benefits from More Supports.** Table 6 shows that existing the TTT-free FSS method encounters performance saturation when the support data reaches 15 shots. In contrast, our ISA method benefits from more support shots, reaching 74.6 mIoU with 30-shot supports.

Table 7: Results of applying our method to other FSS/CD-FSS methods.

| | PANet | + Ours | FPTrans | + Ours | DR-Adapter | + Ours | PerSAM | + Ours |
|---|---|---|---|---|---|---|---|---|
| mIoU | 55.1 | 61.8 | 66.3 | 71.4 | 65.4 | 70.9 | 64.5 | 72.9 |

Table 8: Comparisons with related methods.

| | DG-based Methods | | | TTT-based Methods | | SAM-based Methods | | |
|---|---|---|---|---|---|---|---|---|
| | Mixstyle | DSU | NP | TTT | Tent | PerSAM | Matcher | Ours |
| mIoU | 63.8 | 64.2 | 64.5 | 63.1 | 63.3 | 64.5 | 64.2 | 70.3 |

**Fast ISA.** Based on our experimental analysis, we further propose a fast ISA method. Specifically, we replace ISI by directly selecting the "`layer3.5.conv1`" and the last convolutional layers as the trainable layers. Additionally, we replace PSA with a $(2 \rightarrow 3 \rightarrow 4)$-based progressive training strategy, and randomly select only one training pair for each pseudo query data. The proposed fast ISA method achieves 3.3 FPS and 70.0 mIoU, with a considerable improvement on the running speed and a marginal performance drop compared with the original ISA method.

**Generalized to Other Methods.** Our ISA method is general and can be applied to other FSS methods to address cross-domain few-shot semantic segmentation. As shown in Table 7, when equipped with our ISA method, two FSS methods, PANet (Wang et al., 2019) and FPTrans (Zhang et al., 2022), achieve substantially better performance on CD-FSS. Our method can further improve existing CD-FSS methods, as evidenced by the 5.5 mIoU improvement on DR-Adapter (Su et al., 2024a) when combined with our ISA method. Our method can also adapt the powerful SAM-based method PerSAM (Zhang et al., 2024a) for CD-FSS, achieving remarkable 72.9 mIoU.

**Comparison with More Related Methods.** Table 8 compares our method with domain generalization (Mixstyle (Zhou et al., 2021), DSU (Li et al., 2022), and NP (Fan et al., 2022b)), test-time training (TTT (Sun et al., 2020b) and Tent (Wang et al., 2020)), and SAM-based methods (PerSAM (Zhang et al., 2024a) and Matcher (Liu et al., 2024)) to further demonstrate the superiority of our approach. Our method significantly outperforms other methods. Recently, IFA (Nie et al., 2024) sets a new SOTA on CD-FSS benchmarks, but they adopt a distinct evaluation protocol. For a fair comparison, we adopt their code and evaluation protocol to implement and evaluate our method. When combined with their baseline model, our method achieves 72.8 mIoU, surpassing their reported 71.4 mIoU.

**Discussions on Foundation Model-based Methods.** Foundation models, such as SAM (Kirillov et al., 2023) and CLIP (Radford et al., 2021), are typically trained on large-scale web-collected data, resulting in excellent generalization on natural images. However, due to data domain limitations, they often underperform in unseen domains like medical images, remote sensing images, or industrial images. Additionally, foundation models are typically built on large backbone models, leading to computation, deployment, and storage challenges. In contrast, our method is specifically designed to address generalization to novel classes and unseen domains, featuring lightweight computation, a simple network architecture, few model parameters, and easy deployment. Our method is general and can flexibly equip foundation models to address domain shifts, evidenced in Table 7.

## 5 CONCLUSION

In this paper, we address the domain shifts problem in few-shot scenarios by introducing a novel Informative Structure Adaptation (ISA) method for cross-domain few-shot segmentation (CD-FSS). Our Informative Structure Identification (ISI) module adaptively identifies domain-specific model structures by measuring parameter importance with a novel structure Fisher score in a data-dependent manner. Furthermore, we propose the Progressive Structure Adaptation (PSA) module to progressively adapt the selected informative model structures during inference, utilizing hierarchically constructed training samples with an increasing number of support shots. The ISA method combines these strategies to effectively address domain shifts in CD-FSS, and equips existing few-shot segmentation models with flexible adaptation capabilities for new domains, eliminating the need for redesigning or retraining CD-FSS models on base data. Extensive experiments demonstrate the effectiveness of our method in cross-domain few-shot segmentation.

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
