# OpenReview forum: "Adapting Informative Structures for Cross-Domain Few-Shot Segmentation"
_ICLR.cc/2025/Conference — ICLR 2025 Conference Withdrawn Submission_

### Official Review · Reviewer_mcHd · 2024-10-26

**Soundness:** 3
**Presentation:** 3
**Contribution:** 4
**Rating:** 5
**Confidence:** 4

**Summary:**

The paper introduces the Informative Structure Adaptation (ISA) method for Cross-Domain Few-Shot Segmentation (CD-FSS). This approach tackles the challenge of domain shifts in few-shot segmentation by adaptively identifying and incrementally adjusting model structures during inference. The Informative Structure Identification (ISI) module leverages a novel structural Fisher score to select domain-specific model structures, while the Progressive Structure Adaptation (PSA) module fine-tunes these structures with a progressively increasing number of support shots. Extensive experiments across multiple benchmarks confirm the effectiveness of the ISA method.

**Strengths:**

1. The proposed ISA is interesting for adapting model structures to new domains using few-shot labeled support samples.
2. Extensive experiments have been conducted to demonstrate the effectiveness of the proposed method.
3. The paper is well organized.

**Weaknesses:**

1. The proposed Informative Structure Identification (ISI) module, while innovative, appears to be tailored towards identifying informative structure parameters within CNN-based architectures. However, the manuscript does not sufficiently address how this approach could be extended to Transformer-based models, which have become increasingly prevalent in various computer vision tasks, including semantic segmentation.
2. The Informative Structure Identification (ISI) method seems to be close to the domain of Neural Architecture Search (NAS). However, the manuscript appears to lack a comprehensive discussion on NAS in both the related work and the experimental sections.
3. From Table 3, we can observe the computation cost is a significant drawback of the proposed method,  as reporting a substantial running speed decrease from 16.5 FPS to 1 FPS.  It is a major concern for applications requiring real-time processing capabilities. Additionally, the introduction of the EFI computation significantly increases the computational requirement, potentially making the method impractical for large-scale or time-sensitive tasks.

**Questions:**

My major questions are presented in the weaknesses, looking forward to your response.

---

### Official Review · Reviewer_iD2z · 2024-10-30

**Soundness:** 3
**Presentation:** 3
**Contribution:** 3
**Rating:** 5
**Confidence:** 4

**Summary:**

The paper proposes that the ISA method solves the domain transfer problem by progressively optimising the model structure and parameters, and makes full use of the limited few-sample data to improve the inference performance.

**Strengths:**

1. Embedded use of Fisher information to fully optimize and adjust the model structure
2. Validating the importance of progressive training in gradually solving domain shift

**Weaknesses:**

1. There is not a complete process diagram illustrating what each part does, and the relationships and roles between two parts require many readings of the paper to make the connection.
2. In the ISI phase, why can Fisher information be used as an indicator for tuning layers and parameters, and is there any empirical, theoretical, or experimental evidence for this?
3. Why can Fisher information be used as an indicator for ADJUSTMENT Layer and parameter optimization in the ISI phase? Is there any relevant experience, theoretical support or experimental results to prove this? Specifically, how can the relationship between Fisher information and model performance be derived from existing studies? How is this indicator compared with existing studies to show its superiority?
4. Specifically, the PSA stage needs to clarify the criteria and rationale for sample selection. Is the sample selection strategy based on the distribution of Fisher information, gradient changes, or through some heuristic rule? In addition, the triggering conditions for entering the sample-increasing inference phase need to be identified, e.g., is the expansion of samples started only after a certain degree of loss convergence is reached, or is it based on the dynamic judgement of the model's performance on the existing samples?
5. In the present experimental results, there is only a small increase in performance, and this marginal improvement is likely to be of limited value in practical applications. It is therefore difficult to fully justify the significant superiority of the methodology used on this basis alone. This requires a more in-depth analysis to explain the conditions under which such an improvement is reasonable and meaningful, and to ensure that such marginal improvements are not merely coincidental.
6. Currently there is a lack of sufficient experimental evidence to fully demonstrate that this metric based on Fisher information outperforms other methods, such as gradient-based methods, in all cases. More experimental comparisons are needed to determine whether the introduction of Fisher information actually improves model performance under different tasks or scenarios, and whether it is more generalizable and robust compared to gradient-based methods.
7. The PSA phase of the experiment has not been fully analyzed in comparison to existing sample conditioning strategies. This makes it difficult to judge whether there are unique advantages or improvements in strategies for sample selection in this phase.
8. The association between sample selection strategies and feature information needs to be explored further. Are all Fisher information equally important in sample selection? How does Fisher information guide the process of sample selection, and is there a difference in the role of Fisher information at different stages (e.g., pre-training vs. post-training)?

**Questions:**

Please see the weaknesses.

---

### Official Review · Reviewer_qatP · 2024-10-31

**Soundness:** 3
**Presentation:** 2
**Contribution:** 2
**Rating:** 5
**Confidence:** 4

**Summary:**

This paper tackles cross-domain few-shot segmentation (CD-FSS), a domain-shift scenario that aims to use seen-category supervised models to segment unseen categories. This paper proposes Informative Structure Adaptation, short for ISA, a test-time training, data-dependent strategy that addresses CD-FSS tasks. Specifically, given a test scenario / domain (e.g., medical images), ISA starts with identifies domain-aware model structures (e.g., layers) and fine-tunes these selected structures with few-shot data. ISA showcases performance gains over four CD-FSS experimental setups and verify its effectiveness.

**Strengths:**

- Interesting perspective and clear motivation: training different layers for different datasets. This design stems from the fact that different datasets have different characteristics, thus, simply training the whole model for all datasets is not suitable.

- Substantial experiments on proposed method: conduct a series experiments on ISI and PSA.

- The presentation is clear and the paper is well-written.

**Weaknesses:**

- Unclear training time consumption. As ISA includes another stage of identifying domain-aware model structures, while other CD-FSS approaches do not involve, authors are suggested to include more comparisons on FLOPs or test time consumption. If ISA sacrifices time for performance, authors are suggested to acknowledge this as a limitation. Also, can authors specify how many epochs ISA goes through when performing test-time training?

- Unclear comparison with previous approaches. In line 514-517, I notice that this paper mentions ISA adopts a different evaluation setup distinct from IFA [A], and when using IFA evaluations, ISA surpasses IFA (72.8 v.s. 71.4), which seems an average 5-shot result. Can authors clarify major difference between two evaluation setups, and also provide full comparisons (on 4 datasets, 1-shot and 5-shot), with IFA codebase?

[A] Cross-Domain Few-Shot Segmentation via Iterative Support-Query Correspondence Mining. CVPR 2024.

**Questions:**

Have no further questions so far. Please see weakness 1 and 2.

---

### Official Review · Reviewer_KbwA · 2024-11-01

**Soundness:** 2
**Presentation:** 3
**Contribution:** 2
**Rating:** 5
**Confidence:** 3

**Summary:**

This paper proposes CD-FSS, a cross-domain few-shot segmentation method.
CD-FSS solves two drawbacks of existing methods:
1.  directly applying pretrained CD-FSS models to unseen domains is often suboptimal due to their limited coverage of domain diversity by fixed parameters trained on source domains;
2. adjusting hand-selected model parameters, such as test-time training, typically neglects the distinct domain gaps and characteristics of target domains.

**Strengths:**

1. The proposed technique is not complicated and easy to follow.
2. The proposed modules can be easily integrated in existing methods and improve their performance (Table 7).

**Weaknesses:**

This is an intuitive method, but the motivation seems weak, making the proposed method seems incremental.

W1&Q1: The authors claim that ``adjusting hand-selected model parameters, such as test-time training, typically neglects the distinct domain gaps and characteristics of target domains." Actually, online Test-Time Adaptation (OTTA) and Test-Time Instance Adaptation (TTIA) are two widely studied settings for Test-Time Training (TTT). These techniques can adapt a model to any single unlabeled test image, and goes far beyond adjusting hand-selected parameters [1]. As one of the most important motivations for this article, it is not convinsing enough, I think the author lacks broader literature research.

> [1] A Comprehensive Survey on Test-Time Adaptation under Distribution Shifts.

W2&Q2: Using Fisher information to find important parameters and adjust them seems to be a common idea in the field of machine learning, such as the author mentioned, it is also applied in continual learning. So where is the necessity of applying this technology in few-shot segmentation tasks? In other words, how does the author come up with the idea of ​​applying this technique to this task? The author lacks some analysis on this key point.

W3&Q3: The motivation of progressive structure adaptation (3.3-3.4) is not clear.  Why progressively increasing support samples can deal with domain shift problems ?

Q4: The author uses FIM to find important parameters. I would like to know if there are any rules in the location and importance of these important parameters? (For example, certain parameters are always important in all situations)

**Questions:**

See Weakness Q1-Q4

---

### Official Review · Reviewer_2HyL · 2024-11-03

**Soundness:** 2
**Presentation:** 3
**Contribution:** 2
**Rating:** 3
**Confidence:** 5

**Summary:**

This paper propose using the structure Fisher score to measure parameter importance and adaptively identify domain-specific model structures. To optimize model adaptation and minimize the risk of over-fitting, they also introduce the concept of sustainable learning, utilizing hierarchically constructed training samples that gradually increase from fewer to more support samples. This enhances the flexible adaptability of few-shot segmentation models. The method achieves outstanding performance across multiple benchmark datasets.

**Strengths:**

1.	The writing is fluent and clear.
2.	Using the Fisher score for parameter selection in deep learning is very meaningful, as it provides a reference metric for layer selection in pyramid-like network structures.
3.	Introducing the concept of progressive learning in the context of few-shot tasks with different shot settings is very interesting, as it provides researchers with additional perspectives to consider.

**Weaknesses:**

1.	Feature selection techniques are very common in machine learning and also represent a standard paradigm. While using the Fisher score to measure the importance of neural network parameters is beneficial, it remains a straightforward application and lacks innovation.
2.	Although the authors' model achieves excellent results on multiple datasets, it is not state-of-the-art, and there is a lack of corresponding model comparisons in Table 1, such as IFANet (CVPR 2024).
3.	Although the authors provide a comparison of the trainable layer information structure distribution in Figure 3, there is a lack of detailed explanation regarding what the input 'x' and output 'y' represent, along with the corresponding spatial dimensions, which is confusing. For example, does 'x' refer to the feature map of a single layer or a specific convolutional parameter? Does 'y' refer to the corresponding logit or something else? However, this lack of clarity detracts from the overall understanding of the results.
4.	Table 4 presents informative structure identification (ISI) with different structure Fisher scores. It would be beneficial to supplement this with the selection of different informative structures using (TOP-K) or the Fisher scores that are discarded, to more intuitively measure the differences and effectiveness of this method.
5.	The authors used support images as pseudo-query images to form support-query (pseudo-support) image pairs, and then they progressively increased the number of support samples to construct training pairs. However, the authors lack an explanation of why support images are used as pseudo-query images instead of directly using query images. Additionally, are the support and query (pseudo-support) images the same during training?


[1]  IFANet: Cross-Domain Few-Shot Segmentation via Iterative Support-Query Correspondence Mining

**Questions:**

The questions raised in this section are the same as the weaknesses outlined above.

---

### Official Review · Reviewer_o4u4 · 2024-11-04

**Soundness:** 3
**Presentation:** 3
**Contribution:** 2
**Rating:** 6
**Confidence:** 3

**Summary:**

This paper presents a novel method, "Informative Structure Adaptation (ISA)," aimed at tackling the Cross-Domain Few-Shot Segmentation (CD-FSS) problem. The central concept is to learn domain-specific features from a limited number of annotated support samples, adaptively identifying and gradually adjusting key structures within the model to bridge the domain gap in new domains. Specifically, the method introduces two main modules: the Informative Structure Identification (ISI) module, which employs a structural Fisher score to dynamically pinpoint model structures sensitive to the target domain, and the Progressive Structure Adaptation (PSA) module, which incrementally adapts the model by leveraging additional support samples to reduce overfitting risk and fully utilize limited support data. ISA is model-agnostic and can be readily integrated into existing few-shot segmentation methods to enable cross-domain adaptation without requiring a redesign or retraining of the CD-FSS model on the base data. Extensive experiments demonstrate the effectiveness and superior performance of this method across multiple CD-FSS benchmarks.

**Strengths:**

- This paper introduces an innovative "Informative Structure Adaptation" (ISA) method to address the CD-FSS challenge. ISA learns domain-specific features from limited annotated support samples, adaptively identifying and adjusting key model structures to mitigate the domain gap. The method's model-agnostic design enables straightforward integration into existing few-shot segmentation frameworks for cross-domain adaptation without the need for a redesign or retraining on the base data.
- ISA consists of two pivotal modules: Informative Structure Identification (ISI) and Progressive Structure Adaptation (PSA). ISI utilizes a novel Structural Fisher Score to dynamically recognize model components sensitive to the target domain, while PSA incrementally adapts the model by leveraging additional support samples, thereby balancing the risk of overfitting with the effective use of limited support data. Extensive experiments confirm the effectiveness and superior performance of ISA across multiple CD-FSS benchmarks.

**Weaknesses:**

- ISA could be more clearly distinguished from existing approaches, particularly other CD-FSS methods like PATNet and RD. Comparisons with traditional Fisher Information-based parameter selection techniques would further clarify ISA's unique advantages for CD-FSS-specific challenges.
- The rationale for selecting specific layers for adaptation could be expanded, especially in low-support scenarios. It may also be beneficial to explore whether PSA (Progressive Structure Adaptation) could dynamically adjust layer selection based on domain properties, further enhancing the model's adaptability.

**Questions:**

Please see the Weaknesses

---

### Note · Authors · 2024-11-14

I have read and agree with the venue's withdrawal policy on behalf of myself and my co-authors.